

# DeepMoney: counterfeit money detection using generative adversarial networks

Toqeer Ali[1], Salman Jan[2], Ahmad Alkhodre[3], Mohammad Nauman[4], Muhammad Amin[4] and Muhammad Shoaib Siddiqui[3]

[1] Faculty of Computer and Information Systems, Islamic University of Madinah, Madinah, Saudi Arabia
[2] Malaysian Institute of Information Technology, University Kuala Lumpur, Kuala Lumpur, Malaysia
[3] Faculty of Computer and Information Systems, Islamic University of Madinah, Madinah, Saudi Arabia
[4] Computer Science, FAST-NUCES, Peshawar, Pakistan

## ABSTRACT

Conventional paper currency and modern electronic currency are two important modes of transactions. In several parts of the world, conventional methodology has clear precedence over its electronic counterpart. However, the identification of forged currency paper notes is now becoming an increasingly crucial problem because of the new and improved tactics employed by counterfeiters. In this paper, a machine assisted system—dubbed DeepMoney—is proposed which has been developed to discriminate fake notes from genuine ones. For this purpose, state-of-the-art models of machine learning called Generative Adversarial Networks (GANs) are employed. GANs use unsupervised learning to train a model that can then be used to perform supervised predictions. This flexibility provides the best of both worlds by allowing unlabelled data to be trained on whilst still making concrete predictions. This technique was applied to Pakistani banknotes. State-of-the-art image processing and feature recognition techniques were used to design the overall approach of a valid input. Augmented samples of images were used in the experiments which show that a high-precision machine can be developed to recognize genuine paper money. An accuracy of 80% has been achieved. The code is available as an open source to allow others to reproduce and build upon the efforts already made.

Corresponding author
Toqeer Ali, toqeer@iu.edu.sa

# INTRODUCTION

Currency is a common system to perform transactions for trading or exchange of goods among people. Various currencies are recognized for trading between nations in foreign exchange markets. The problem with paper currency is that it may be counterfeit. Counterfeit is imitation money which is produced without the legal sanction of the state or government, considered as fraud (*Derrida, 1994*). Anti-counterfeiting measures can be adopted which involve the fine details of the raised intaglio printing on notes which allows non-experts to easily spot forgeries (*Tanaka, Nishiyama & Koyama, 1996*).

With advancement in technology, fakes and forgery rates are increasing. In 2015, almost 70% of the $78 million in counterfeit currency circulating in the U.S was made using

digital printing technologies (*Murakami-Fester, 2016*; *Bartkiewicz et al., 2017*). As reported by Bloomberg news, a 34-year-old hairstylist forged up to $20,000 in counterfeit notes (*Magdaleno, 2016*). Another research carried out by Gallup in Pakistan found out that more than a quarter of the population (26%) have received counterfeit money while buying items from the market. Elsewhere, a raid by Peshawar police yielded fake documents and 0.7 million in counterfeit currency notes. The chairman and managing director of a security printing corporation in Pakistan said that enemy countries were producing counterfeit notes of Pakistani rupee which was being used exclusively for terrorist activities in the country (*Shoaib et al., 2013*; *Taillard, 2018*). The State Bank of Pakistan has carried out various types of campaigns to raise awareness of the security features of bank notes, both directly to its customers and in collaboration with other banks. Bank's efforts to raise awareness amongst the general public involve a media campaign and a mobile application that detects counterfeit notes. The awareness of the security features of bank notes in public may help some people identify counterfeit notes; however, most of the public, especially people who are illiterate, are not able to differentiate between a forged currency note and a genuine one. Also, these features are hard to recognized by human eye or touch when the currency notes are old, dirty, and damaged.

To solve the issues in classifying a currency note as a fake or a genuine note, we have proposed a machine assisted system named DeepMoney. For discriminating fake notes from genuine ones, state-of-the-art models of machine learning called Generative Adversarial Networks (GANs) are employed. GANs use an unsupervised learning to train a model that can then be used to perform supervised predictions. This flexibility provides the best of both worlds by allowing unlabelled data to be trained on whilst still making concrete predictions. This technique is applied to Pakistani banknotes. State-of-the-art image processing and feature recognition techniques were used to design the overall approach of a valid input. The rest of paper is organized as follows. 'Related work' discusses the related work and provides details of various deep learning models in the subject domain. 'Proposed solution' provides details of the proposed solution and how the dataset was developed. 'Results' presents the results and evaluation details of employing models on the dataset. The paper is concluded in 'Conclusion'.

## RELATED WORK

Bearing the aforementioned issues in mind, a number of research solutions have been provided in the past to check the validity of the banknotes (*Thakur & Kaur, 2014*; *Mirza & Nanda, 2012a*; *Chakraborty et al., 2013*; *Prasanthi & Setty, 2015*; *Kang & Lee, 2016*). *Mirza & Nanda (2012b)* offered a solution for a currency verification system using image processing based on the extraction of characteristics. The solution was applied to Indian banknotes. The edge detection and image segmentation were used to make a comparison between the original and the counterfeit notes.

Snehlata et al. presented a UML activity model designed to represent the dynamic aspects for identification of fake currency for Rs 2000 currency note for Indian rupee. They have used class descriptions for real and fake images of the currency for security threads in the

form of strips and apply comparison of block pixels to identify fake and real currency (*Snehlata & Saxena, 2017*).

Singh et al. have presented an image processing-based approach for detecting forged Indian currency. The authors have utilized security thread and latent image, embedded on the note to identify forgery. First the security features are extracted and encoded, then a clustering algorithm, k-means is applied for classification. Then the latent image, segmented via template matching is encoded using HOG descriptor and classified with an SVM model to predict if the note is fake or real *Singh, Ozarde & Abhiram (2018)*.

*Abburu et al. (2017)* proposed a system for automated currency recognition using image processing techniques for accurately identifying both the country of origin and the denomination of a given banknote. However, they do not discriminate between a fake and a real currency note. However, the DeepMoney solution proposed here does not use image processing and differs in many ways.

*Ross et al. (2016)* have proposed a database for detecting counterfeit items using digital fingerprint records which can be used for detecting counterfeit currency note. It takes an image of the authentication region and creates a digital fingerprint of the object. It uses signal processing techniques, such as, FFT of the image to create the digital fingerprint to extract features which are used to compare the fake and real objects.

Kayani presents a bank note processing system based on florescence and phosphorescence detection (*Kayani, 2017*). The illumination source is used to direct light on a note and the sensors measures the florescence and phosphorescence that are used to identify if the note is fake or real. *Micali & Devadas (2017)* proposed a solution for counterfeit prevention using physically unclonable value for unique identification for each currency note.

Phillips has presented a miniature counterfeit detector in his patent (*Phillips, 2018*). It applies multiple test to assure the authenticity of a currency note. Back light illuminators are used for visual inspection of the watermarks, florescent and anti-counterfeiting features. Sensors, such as, magnetic ink sensor, are used to detect the magnetic ink based security features on the note. However, some of the features are detected but with old notes the rate of false negative would be high.

Alicherry has given a digital signature based solution for verifying the authenticity of a currency note and tracking duplicate notes. A digital signature based on the serial number of the currency note is attached to the currency note and people can query the authenticity of the note by sending a photograph of the note to a centralized server for verification and tracking *Alicherry (2017)*.

Before GANs (*Goodfellow et al., 2014*), a neural network approach was used to authenticate banknotes (*Mohamad et al., 2014*). In this research, Generative Models to train a neural network are preferred. Python has also been used to develop and implement a framework for the identification of Pakistani currency.

*Berenguel et al. (2016)* also worked on methods by which to identify genuine and photocopied bank notes. The technique applied was to differentiate the texture between the original and photocopied notes. Other studies *Choi et al. (2010)* also worked on the detection of counterfeit banknotes by using optical coherence tomography (OCT).

To differentiate between genuine and counterfeit notes, the researchers used a three-dimensional imaging security feature according to the FF-OCT system. Their results show that it is possible to recognize original notes with the DeepMoney technique. However, their technique was based on a particular feature of specific banknotes which may prove to be less effective on other notes. This also differs from the DeepMoney perspective of recognizing banknotes. Table 1 elaborates some recent research work regarding counterfeit money.

There have been a few solutions provided which use machine learning techniques. For example, *Hassanpour & Farahabadi (2009)* used Hidden Markov Models for the recognition of banknotes. In the following subsection, we discuss some of the deep learning model that could be utilized in the field of identifying currency forgery and counterfeit currency.

## Deep learning models

Some of the best known deep learning models in comparison to our proposed GANs model on the grounds for selecting the one which will offer the most robust results, are discussed in the following subsections.

## Recurrent neural networks

A number of techniques exist in traditional machine learning and deep learning which allow patterns in sequences and contents to be learned. A recurrent neural network is one such technique. Context sensitive and inherently ordered data is used in this network. The network can operate with both audio and text input. Recurrent neural networks are adept at handling arbitrary length sequence data. This network is a powerful tool which requires the sequence to be contextual. Recurrent neural networks are still very dominant although, in the past, they were very hard to train. However, there is now a solution, Hessian-free optimization, which offers the ability to train recurrent neural networks. The model is depicted in Fig. 1.

## Fully connected neural networks

As its name indicates, a fully connected neural network is one in which all the neurons are connected to the next layer of neurons. There are many layers such as the max pooling and the convolutional layers. The high level perception in these networks uses only one type of layer and these layers are fully connected. These connected layers in the neurons link to all the activations in the previous layers. The activation function may also be calculated by the multiplication of a matrix trailed by a network's 0 set, also known as bias. A fully connected neural network has an input layer, a hidden layer and an output layer as depicted in Fig. 2.

### Convolutional neural network

This type of neural network processes the dimensioned and order data in different ways. Convolutional neural networks (*Krizhevsky, Sutskever & Hinton, 2012*) are learned through the same method as the stochastic gradient descent method traditionally learns. The following are the convolutional neural network layers:

**Table 1   Related work done in the recent years in the field of counterfeit currency detection.**

| Authors | Objective | Method | Limitations | Year |
|---|---|---|---|---|
| *Thakur & Kaur (2014)* | Review of fake currency detection techniques | Survey paper | Not applicable | 2014 |
| *Chakraborty et al. (2013)* | Recent developments in paper currency recognition system | Survey paper | Not applicable | 2013 |
| *Prasanthi & Setty (2015)* | Indian paper currency authentication system | Image processing | Performance is less than machine learning based systems | 2015 |
| *Kang & Lee (2016)* | Fake banknote detection | Multispectral imaging sensors | Feature extraction and classification require high computation | 2016 |
| *Mirza & Nanda (2012a)* | Currency verification | Image processing: edge detection and image segmentation | Only for Indian notes | 2012 |
| *Snehlata & Saxena (2017)* | Fake currency identification | UML activity model using class descriptors | Only for Rs 2000 note of Indian currency | 2017 |
| *Singh, Ozarde & Abhiram (2018)* | Detecting forged Indian currency | Image processing, k-means clustering and SVM as a classifier | Limited to Rs 500 note of Indian currency | 2018 |
| *Abburu et al. (2017)* | Automated currency recognition for identifying country of origin and denomination | Image processing | Cannot detect counterfeit or forgery | 2017 |
| *Ross et al. (2016)* | Database for detecting counterfeit items | Digital fingerprint records using images of security features | Performance is less than machine learning based systems | 2016 |
| *Kayani (2017)* | Bank note processing system | Florescence and phosphorescence detection | Many security features are not detectable using florescence and phosphorescence detection | 2017 |
| *Micali & Devadas (2017)* | Counterfeit prevention | Physically unclonable value for unique identification for each currency note | Needs Internet connection for sending images to centralized server | 2017 |
| *Phillips (2018)* | Miniature counterfeit detector | Back light illuminators are used for visual inspection of the 93 watermarks, florescent and anti-counterfeiting features | Many security features are not detectable using florescence and phosphorescence detection | 2018 |
| *Alicherry (2017)* | Verifying the authenticity of a currency note and tracking duplicate notes | Digital signature based on the serial number of the currency note | Needs Internet connection for sending images to centralized server | 2017 |
| *Berenguel et al. (2016)* | Identify genuine bank notes | Differentiate the texture between the original and photocopied notes using OFT | Accuracy is less than machine learning based systems | 2016 |
| *Choi et al. (2010)* | Counterfeit detection | Characterization of safety feature on banknote with full-field optical coherence tomography | Accuracy is less than machine learning based systems | 2010 |
| *Hassanpour & Farahabadi (2009)* | Paper currency recognition | Machine Learning: Hidden Markov Models | Accuracy is less than the proposed system | 2009 |
| *Mohamad et al. (2014)* | Banknote authentication | Srtificial neural network | Accuracy is less than the proposed system | 2014 |

*Convolutional*  In this layer there is a grid of neurons. Commonly, this grid is rectangular which requires that the previous layer should also take the form of the same rectangular shaped grid. In the convolutional layer, the neurons have the same weight. Each neuron takes the input from the rectangular section with the input coming from the previous layer.

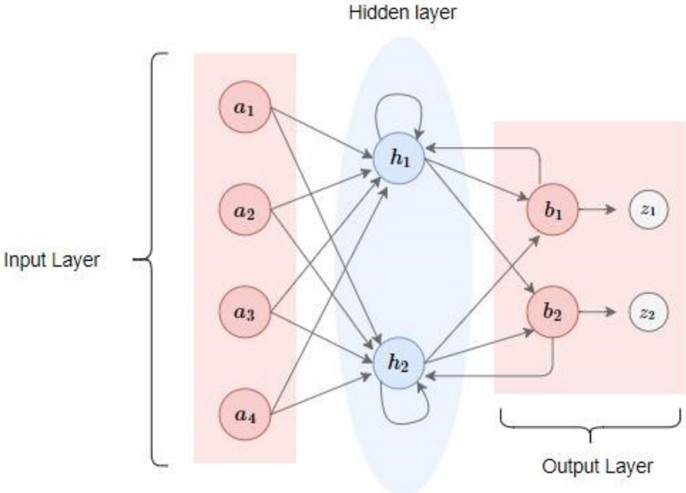

Figure 1   A recurrent neural network.

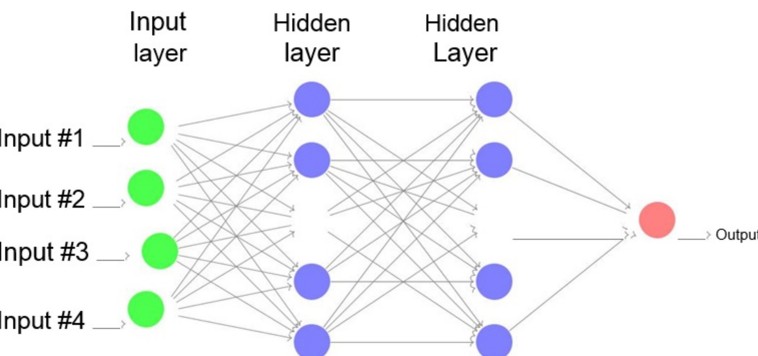

Figure 2   A fully connected neural network.

*Max-pooling*  Pooling layers are present after each convolutional layer. The layer usually grabs tiny rectangular blocks from the convolutional layer and then samples them so that only one output is created. Pooling can be achieved in many ways such as by taking an average or by learning patterns and combinations, for example learning linear associations or the combination of neurons in that small block.

*Autoencoder*  There are many machine learning models but auto-encoder are fairly basic models. They come from a family of neural networks for which the input is same as the output. Auto-encoders compress the latent space representation and then reconstruct the output from the representation. Auto-encoders are used for unsupervised learnings and so are an artificial network. The learning of encoders are efficient enough. Today, auto-encoders are an emerging field of research in numerous areas, such as anomaly detection.

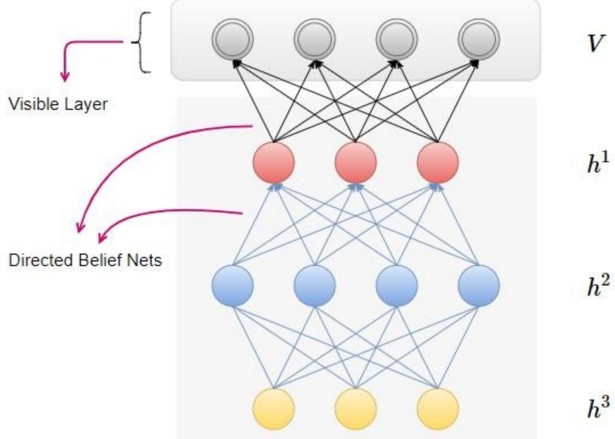

$$P\left(V,h^1,h^2,\ldots,h^l\right) = P\left(V|h^1\right)P\left(h^1|h^2\right)\ldots P\left(h^{l-2}|h^{l-1}\right)P\left(h^{l-1}|h^l\right)$$

**Figure 3   Deep belief networks.**

*Deep belief networks*  Deep belief networks (*Lee et al., 2009*) consist of layers of variables, both latent as well as stochastic. Latent variables are usually composed of values that are binary and known as feature detectors. There are many layers in this network. The top layers consist of connections that are undirected which allows them to form a kind of associative memory. On the other hand, the lower layers are directed links from any previous or top layer. A lower layer represents a data vector. An example of Deep Belief Networks is depicted in Fig. 3:

*Long short-term memory*  The connected neural network, also called the recurrent neural network, works but, when applied on different models, it suffers from two problems. Firstly, it produces a vanishing gradient and, secondly, it is an exploding gradient. LTSM (long short term memory) was invented to solve these issues by introducing a memory unit, which is called a cell, into the network. The cell is used as storage or memory that basically remembers a value for long or possibly short time periods (*Hochreiter & Schmidhuber, 1997*; *Sak, Senior & Beaufays, 2014*; *Warrington & Baddeley, 2017*). LSTM are represented in Fig. 4.

## Problem statement

The existing solutions only work to detect the security features of the notes and at present this is the only viable solution used to tackle this issue. Forged notes are not detectable by the naked eye and this results in financial loss to the general public. Degraded bank notes are also in circulation which may result in usage problems. The solution given in this paper is based on Generative Adversarial Networks (GANs) (*Goodfellow et al., 2014*) that use generative and discriminative models for the recognition of real and counterfeit currency notes. To the best of our knowledge, this is first research being done to detect forged money using GANs. We trained the discriminative model with Pakistani rupee notes while the

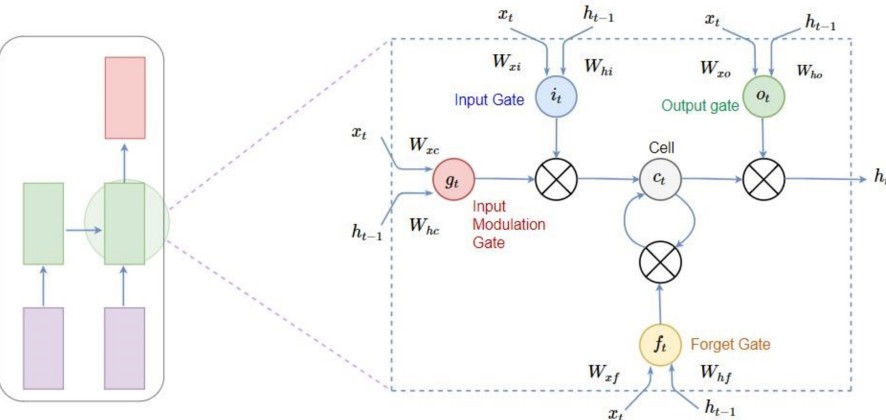

**Figure 4  Long short-term memory.**

generative model G was used to input the counterfeit notes to classify with discriminative model D. The objectives of the studies is twofold:

1. To build a dataset of real and counterfeit Pakistani currency which is not available at present in the public domain;
2. To present, design, and implement DeepMoney, a method to differentiate counterfeit notes from genuine bank notes.

# PROPOSED SOLUTION

In this section, the proposed solution is elaborated and details are provided as to how Generative Adversarial Networks cope with real and fake data.

## Generative adversarial networks

A very new and effective Generative Model is utilized for generative counterfeit samples and for recognizing original data items from the generative ones, known as Generative Adversarial Networks (GANs) (*Goodfellow et al., 2014*; *Goodfellow et al., 2016*; *Radford, Metz & Chintala, 2015*; *Salimans et al., 2016*). GANs can differentiate with maximum accuracy between the fake and real banknotes. GANs are quite interesting phenomena of neural networks that work on two modules. One is called the generative network and the second is known as the discriminator network. Quite promising results have been achieved after employing GANs on the dataset and then subsequently used to classify the real and fake notes. The proposed model is depicted in Fig. 5 wherein the Discriminator Neural Network (D) is trained by providing data from training set (original distribution) and generated data (after perturbation i.e., adding noise from the latent space). The loss functions are updated during training process of each Discriminator's network and the Generator Network (G). After training is completed, the model is able to classify real and fake currency notes with accuracy of 80 percent.

In GANs, the discriminator part works as a classifier for recognizing the images. However, in the learning process, both the generator and discriminator co-ordinate with each other.

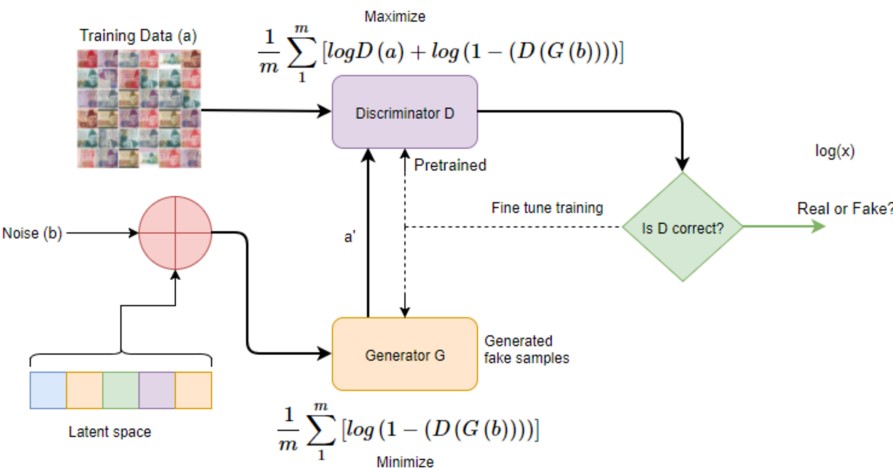

**Figure 5  DeepMoney: proposed counterfeit model.**

A generated image is sent to the discriminator module to classify whether it is a fake or a real image. If it is recognizable, the discriminator will produce its output, if it is not, the module will send it back to the generator to regenerate the image. Based on the feedback received, the generator improves its cast and creates the image. The process continues until both the models are optimal in correctly generating and classifying the same image.

In the case of currency note identification, the basic idea is that there are two models as defined by GANs. The generative model generates fake banknotes and the discriminative model D estimates the probability that whatever data is received from G, it is from training data or it is from generator G itself. To understand the generator's distribution $P_g$ over data ($a$), a priori is defined on input noise variables $P_b(b)$, which then represents a mapping to the banknotes data space as $G(b; \theta_g)$, where $G$ is a differentiable function represented by a multilayer perceptron with parameters $\theta_g$.

According to GANs networks, a second multilayer perceptron was also designed $D(y; \theta_d)$ that outputs a single scalar. $D(a)$ represents the banknotes that came from the data, not from the generator. The discriminator $D$ was trained with real banknotes to increase the probability of correctly recognizing the $Data(a)$ and the sample $b$ was produced from the generative model $G$. The loss or energy function of generative model $G$ can be represented mathematically as:

$$\frac{1}{m} \sum_{1}^{m} [log(1 - (D(G(b))))]$$

and the loss function of the Discriminator can be represented as:

$$\frac{1}{m} \sum_{1}^{m} [logD(a) + log(1 - (D(G(b))))]$$

Figure 5 shows a complete flow of how our DeepMoney process works. Real images are given as input to the discriminator, while for training, communication is performed between the discriminator and the generator.

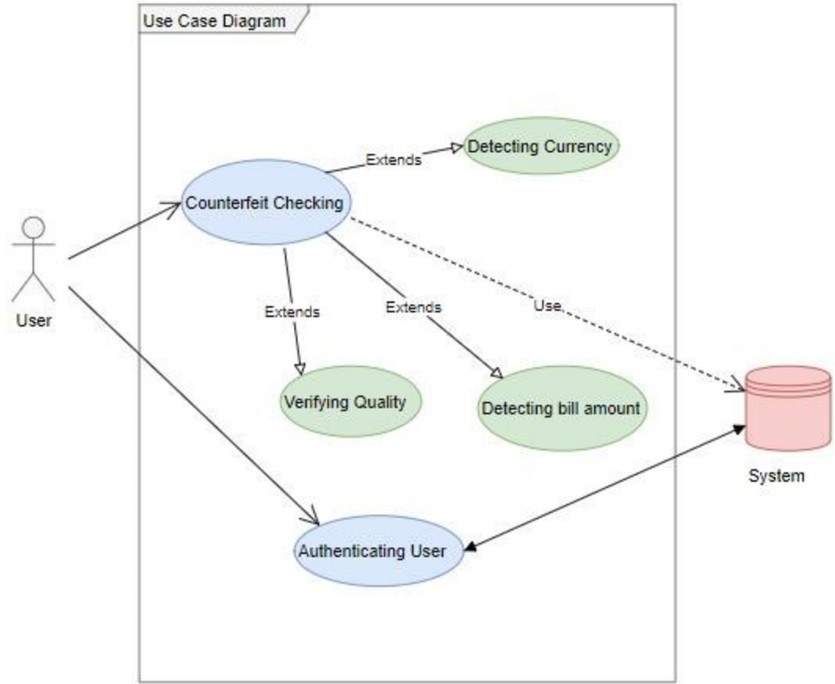

**Figure 6**  **DeepMoney currency bill verification components.**

Different components have been built to perform the verification of the currency note. Figure 6 shows the basic structure of the DeepMoney architecture, its actors, and the way they interact with each other. There is a single actor here, which is shown as the user. The user can input an image and request the authentication of the currency note. The system will respond through various other functions, as shown in Fig. 6. External systems may be used to receive assistance from the sensors if necessary. As DeepMoney progresses, other more appropriate features may be found and the use of these initial features may be minimized and other more versatile features may evolve. In this case, the 'input image' will remain the same but the dependent functionalities may change.

## Data preparation and augmentation

Data collection and its preprocessing is carried out before training neural networks or deep learning models for subsequent recognition tasks. The following subsections provide the necessary details of how data is prepared and augmented with image datasets through the following function:

**datagen = ImageDataGenerator()**

Taking pictures and building a dataset is a laborious and time consuming task. An API specially designed for creating augmented image data is used which takes less time to augment the data and reduces memory overhead. ImageDataGenerator augments the data and when data has been created and configured then it is time to fit the data by calling the fit() function. Once the function is called on the data generator it is then passed in the training dataset.

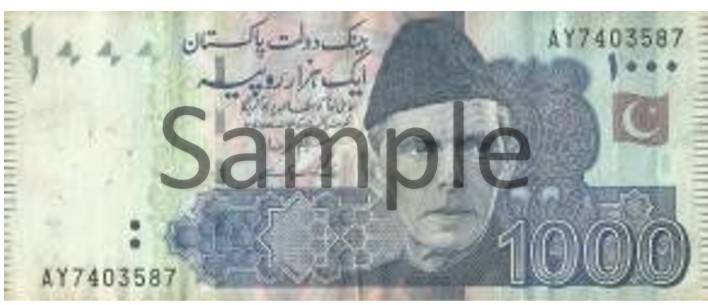

**Figure 7** Input image.

**datagen:fit(train)**

Data generator is an iterator and it returns the batches of image back when they have been requested.

$X_{batch}; Y_{batch} = datagen : flow(train; train; batchsize = 32)$

Finally, fitgenerator() is called with arguments of the desired length of epochs on which to train the data. The other one is the data generator.

**fitgenerator (datagen; samplesPerEpoch = len (train) ; epochs = 100)**

There are many ways to configure the data preparation and its augmentation. These include, feature-wise standardization, ZCA whitening, random rotation, shifts, shear and ips, dimension reordering, saving augmented images to disk. All the functions/augmentation are performed by calling the ImageDataGenerator function. As a result of providing the input image as depicted in Fig. 7, the function ImageDataGenerator produces augmented data through incorporating the aforementioned features as shown in Fig. 8.

## Experimental setup

The experiments are carried out on a GPU machine in FAST University with the following configuration. Keras is configured with Theano as backend. From the Keras library, the packages, Dense, Dropout, Activation, and Flatten layers are imported. The Dense layer is used to define filters with model parameters to identify various features of the currency notes. The Dropout layer is used to address overfitting and in order to ignore some of the features which are not contributing in identifying the actual features of the notes. The Activation function, "Relu" is used in order to represent the learned information in ranges of 0 and 1. The number of epochs and batch size is set to 100 and 32, respectively.

To use the ImageDataGenerator function, Keras was utilized beside configuring scipy, numpy, h5py and pyyaml. Furthermore, a number of parameters were set for achieving augmentation which included: samplewise center, featurewise std normalization, ZCA Pesilon, ZCA whitening, rotation range, width shift range, height shift range, shear range, zoom range, channel shift, horizontal, vertical, and rescale. These are set to either, 0, 1 or in float values for the required changes. After specifying the parameters and storing the same in a datagen variable, the images were imported.

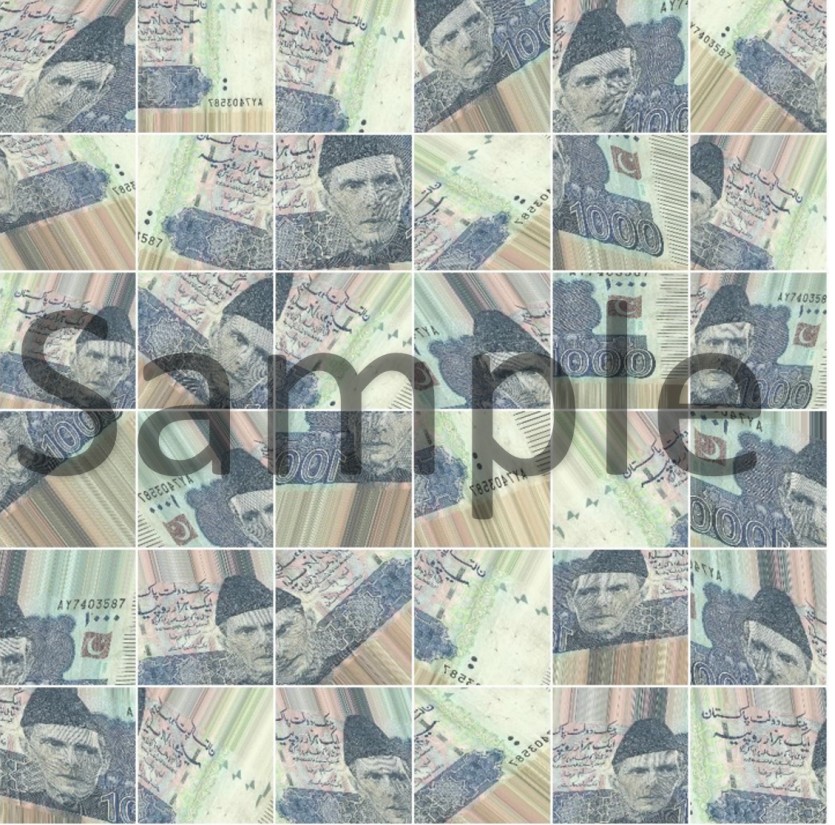

**Figure 8  Output images.**

As specified by the 36 iterations, 36 images of the Bank Note with changes that were specified in the datagen will be produced and will be stored in the folder called preview.

## Implementation

Comprehensive details are provided as to how the environment was set up for executing this project of GANs on the dataset. The first step involved the setup and configuration of the requirements that would help to execute the code. After running certain required scripts, GAN was trained on the DeepMoney dataset. Both the discriminator and the generator used a learning rate of 0.0002. The generator updated this twice for each update of the discriminator. This actually avoids the small discriminator loss. In the next step, results are produced and saved in PNG format.

The class diagram in Fig. 9 represents the major classes of DeepMoney. As can be seen, the main controller class is InputImage and every other class is developed from that class. InputImage reads an image and passes that image to one of those classes, or to multiple classes, according to the requirement. The respective classes will then call their functions that may call another function of different classes.

The sequence diagram in Fig. 10 shows the functions and their activities after they have been executed. As discussed earlier, functions may or may not call other functions of

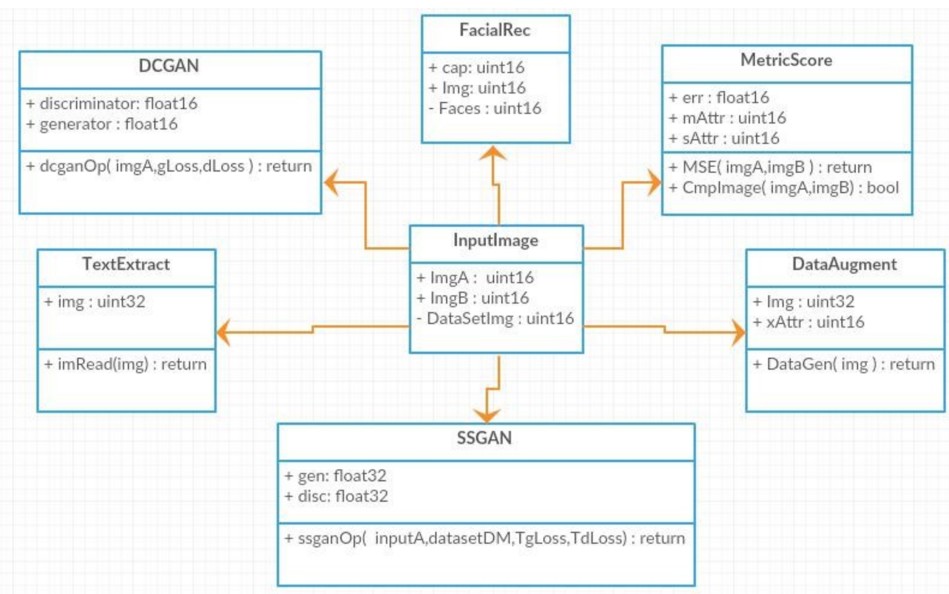

**Figure 9   Class Diagram for DeepMoney Architecture.**

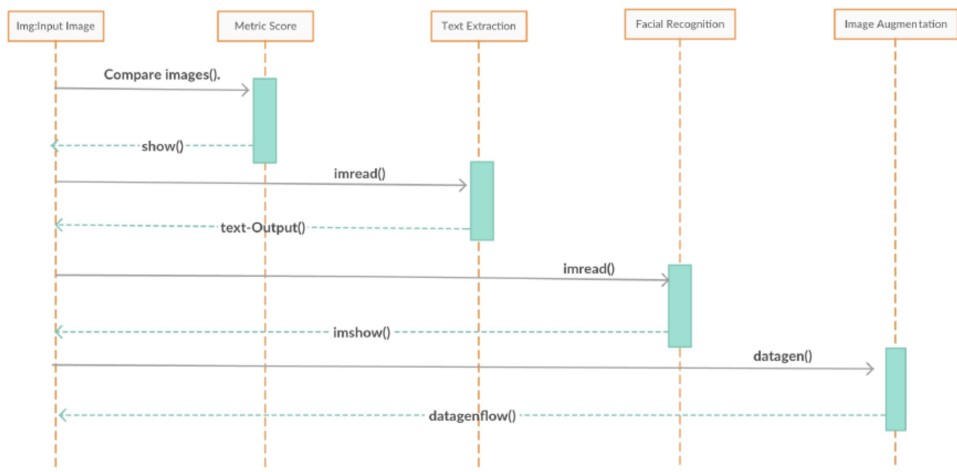

**Figure 10   DeepMoney sequence diagram.**

different classes if the task required is beyond their scope. The functions of other classes are called through instances of their respective class. The called function then returns the result to the parent function which, in turn, processes the user request.

The algorithm for the GANs is provided in Fig. 11, while the algorithm for generating images is provide in Fig. 12 and the algorithm for augmentation is provided in Fig. 13.

```
                    ┌─ Algorithm for Generative Adversarial Network ─┐
1   for <number of training iterations> do
2   for <k steps> do
3   Sample minibatch of m noise samples fb1..., bm g from noise prior Pg(b)
4   Sample minibatch of m examples fa1, .... , amg from data
5   generating distribution pdata(a)
6   Update the discriminator by ascending it's stochastic gradient
7   end for
8   sample minibatch of m noise samples fb1, ... , bmg from noise prior
9   Pg(b) update the generator by descending it's stochastic gradient
10  end for
```

**Figure 11**    **Algorithm for generating currency notes.**

```
                           ┌─ Algorithm for Generating Images ─┐
1       x = img to array (img)
2       creating a Numpy array with shape (3, 150, 150)
3       x        = x:reshape ((1) + x:shape)
4       converting to a Numpy array with shape (1, 3, 150, 150)
5       Loop1 runs for 36 times
6       for each iteration
7       datagen:flow ()
8       #function was used with x being given the numpy array for the input image
9       save to dir, the directory to save output
10      save pre x, the pre x for the names of the images and
11      save format, the image format as input
```

**Figure 12**    **Algorithm for generating currency notes.**

```
                              ┌─ Augmentation Algorithm ─┐
1       Procedure from keras.preprocessing.image(import ImageData-Generator)
2       datagen   ImageDataGenerator rotRng,wtSft,htSft,zoomRng,hFlp,
3       Mde x        = x:reshape ((1) + x:shape)
4       while r6= 0 do
5       img   loadImage
6       x    imgT oArr
7       x    x:reshape
8       i    0
9       end while
10      for <batch in datagen.flow> do
11      <i+=1>
12      end for
13      if i > 36
14      break
15      end procedure
```

**Figure 13**    **Algorithm for generating currency notes.**

# RESULTS

This section elaborates the results obtained through the experiments conducted on the DeepMoney dataset as depicted in Table 2. Moreover, a model evaluation is presented with the confusion matrix, supervised loss function, the generator and discriminator respective losses and finally the classification accuracy.

## Area Under the Curve (AUC)

The trained model was able to achieve 80% accuracy in correctly classifying data. The Area Under the Curve (AUC)/Receiver Operating Characteristic (ROC) is presented in Fig. 14.

## Confusion matrix

The confusion matrix consists of three classes along the $x$ and $y$ axis. Each row of the matrix represents each class along with the $y$-axis which shows its comparison with all the remaining classes. The classes belong to the DeepMoney dataset having images of Rs. 50,

**Table 2 Experimental results.**

| No. of Folds | Accuracy | Sensitivity | Specificity | Precision | Kappa | F1 Score | ROC/AUC |
|---|---|---|---|---|---|---|---|
| 5 | 76.8761% | 0.7301 | 0.9771 | 0.7518 | 0.1283 | 0.7567 | 0.7598 |
| 10 | 79.5420% | 0.7356 | 0.9783 | 0.7540 | 0.1854 | 0.7632 | 0.7745 |
| 15 | 73.9300% | 0.7393 | 0.9798 | 0.7387 | 0.1767 | 0.7689 | 0.7965 |

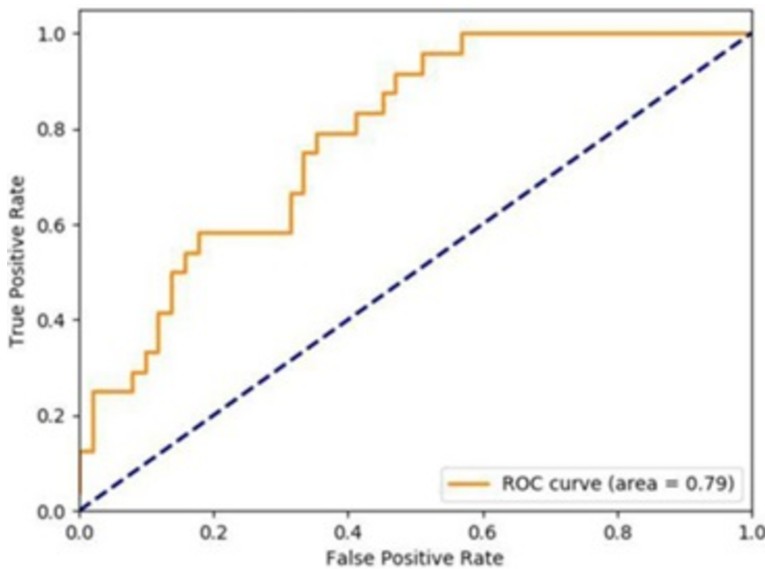

**Figure 14 DeepMoney area under curve.**

Rs. 500, and Rs. 1000. Each class is assigned a unique density color. High intensity shades represent no confusion between classes while the lower intensity band represents confusion rates between the classes. In Fig. 15 the diagonal showing 1′s indicate that the confusion matrix calculation is near perfection.

## Supervised loss

Cross-entropy was minimized for the multi-class softmax and for that purpose the labels were adjusted. A label mask in code was used to ignore the images that are unlabeled for the SS-learning problem. The TensorFlow graph visualization for the (percentage vs No. Epochs) are shown in Figs. 16–18.

## Total discriminator loss (TD Loss)

Using the objective function of GANs, the Total Discriminator loss (TD-Loss) is brought down to 0.5. This helped to make the Discriminator more efficient in classification. The Tensorflow Graph Visualization below is pitted against (PercentageLoss No. Epochs) in Fig. 16.

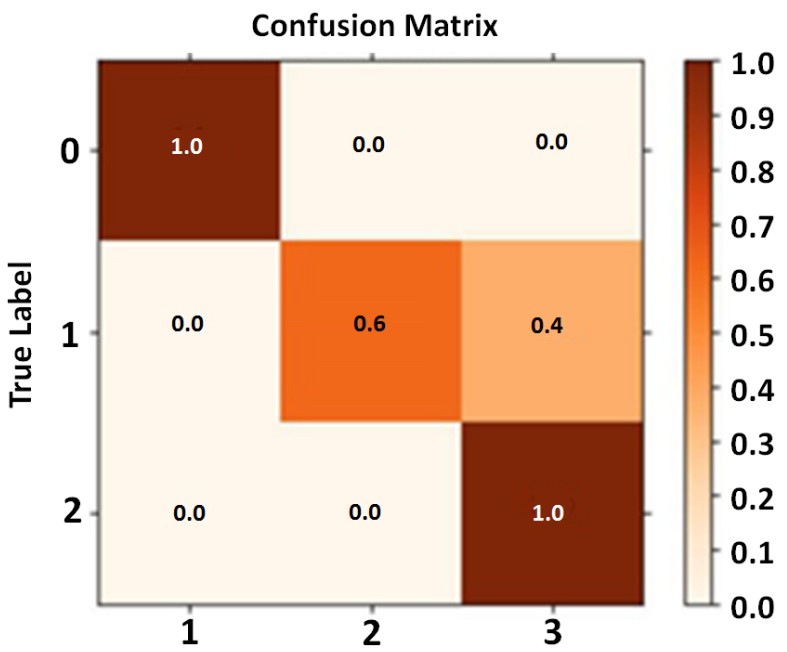

**Figure 15  Confusion matrix calculation.**

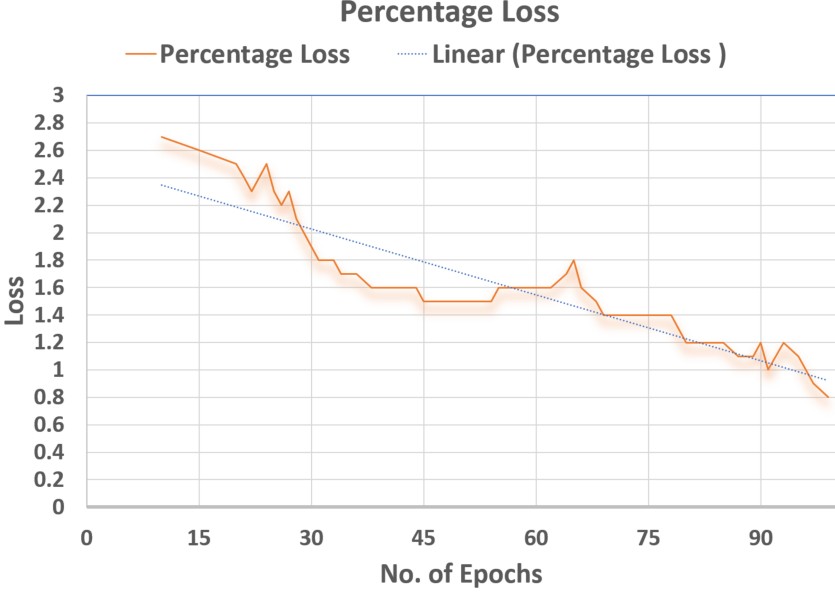

**Figure 16  Discriminator loss graph visualization.**

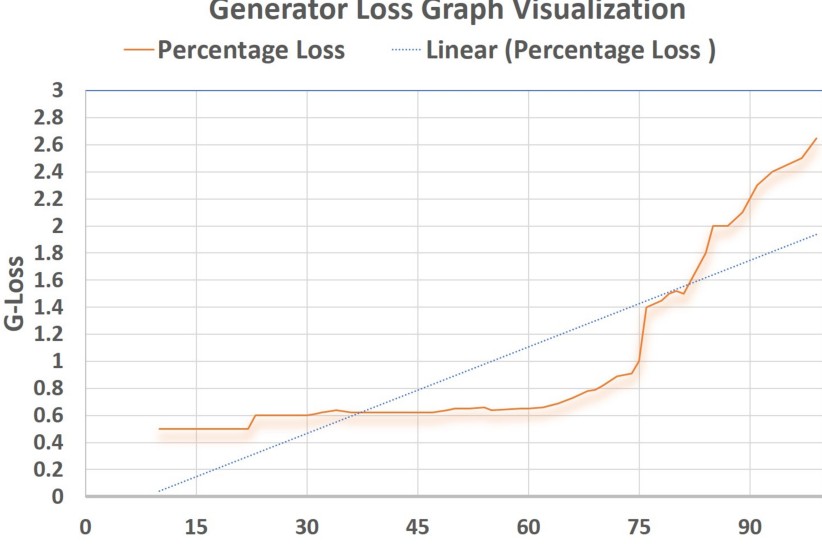

**Figure 17** **Generator loss graph visualization.**

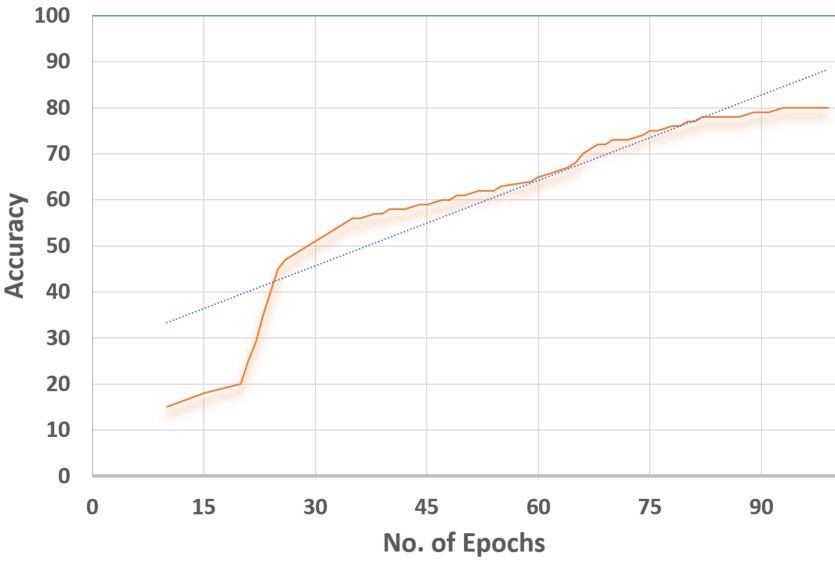

**Figure 18** **Classification accuracy graph visualization.**

## Generator loss (G Loss)

Using the Objective function of GANs, the Generator's loss (G loss) was brought up to (2.65), which can be seen in Fig. 17. This helped to make the Generator more robust in training the Discriminator. The Tensorflow Graph Visualization is visualised against (Percentage-Loss vs No. Epochs).

## Classification accuracy

The Discriminator is trained to classify the unseen images and the efficiency of the same is measured in terms of correct classification. As can be seen in Fig. 18, a remarkable classification accuracy (80%) was achieved, which was made possible by training a Discriminator through DeepMoney dataset and Generator images. The Tensorflow Graph Visualization is depicted in Fig. 18 (Percentage Accuracy vs. No. Epochs).

## CONCLUSION

Implementation of GANs models in the domain of computer vision has been proven to be effective. Upon testing in the currency originality, the discriminator model is seen as a viable contender in the classifier area. Additionally, classifier should be sufficiently trained with enough images by means of the real dataset and the generator module, especially in the Semi Supervised Generative Adversarial Networks (SSGANs). The dataset alteration can be performed in various other parametric tweaking with the Keras framework. 80% accuracy is achieved by the proposed GANs framework for counterfeit money detection; however, there is still room for improvement. As new generative models are created in the machine learning domain, DeepMoney can be tested upon these to achieve better and more effective results. Multi-class classifiers can be made good enough to result in improved accuracy. Classification is regarded to be at the forefront of machine learning, therefore, a better multi-class classifier would yield an even better retrospective score for the model.

### Funding
The authors received no funding for this work.

### Competing Interests
The authors declare there are no competing interests.

### Author Contributions
- Toqeer Ali conceived and designed the experiments, contributed reagents/materials/-analysis tools, approved the final draft.
- Salman Jan conceived and designed the experiments, performed the experiments, prepared figures and/or tables, performed the computation work, authored or reviewed drafts of the paper, approved the final draft.
- Ahmad Alkhodre contributed reagents/materials/analysis tools, authored or reviewed drafts of the paper, proof-reading, other support.
- Mohammad Nauman performed the experiments.
- Muhammad Amin analyzed the data, performed the computation work.
- Muhammad Shoaib Siddiqui dataset and Code.

### Data Availability
Ali, Toqeer; Jan, Salman (2019): DeepMoney: Counterfeit Money Detection Using Generative Adversarial Networks. figshare. Dataset. https://doi.org/10.6084/m9.figshare.9164510.v3.

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
