# Peer review of "DeepMoney: counterfeit money detection using generative adversarial networks"

_PeerJ Computer Science, doi:10.7717/peerj-cs.216_

## Round 0.1 · original submission · Major Revisions

Please respond to all referees' concerns, including the language issues.

·

Basic reporting

Paper is well written, English is good. References are properly marked. Fig 5 (picture) is bit crowded, could have been better structured so could be easier to follow.

Experimental design

Good, Question/problem well defined. Details are good enough.

Validity of the findings

Impact is not fully assessed, meaning the loss of 20%, under what conditions the accuracy of detection is failing is not clearly articulated.
Conclusions are well stated.

Additional comments

Overall good.

Reviewer 2 ·

Basic reporting

Literature Review / Related Work is not adequate. In fact it is merged with the Introduction section. Although several techniques have been proposed to this problem for the last 25 years. Moreover following works have been presented on this area since last 2 years in various conferences, and journals, which authors did not refer:

Verifying Authenticity of Currency and Tracking Duplicates by M Alicherry - 2017

Database for detecting counterfeit items using digital fingerprint records by DJ Ross, BJ Elmenhurst, M Tocci, J Forbes-

Bank note processing system having a combined florescence and phosphorescence detection system by S Kayani

Image Processing Based Detection of Counterfeit Indian Bank Notes by M Singh, P Ozarde, K Abhiram

Currency Recognition System Using Image Processing by SS Rajebhosale, DS Gujarathi, SV Nikam, PP Gogte

Counterfeit prevention by S Micali, S Devadas

MINIATURIZED COUNTERFEIT DETECTOR by R Phillips

Identification of Fake Currency: A Case Study of Indian Scenario by S Snehlata, V Saxena

Only two references are used from last two years which is inadequate.

The related work should compare the use of GAN with other techniques used in solving this problem. Moreover the Discussions on Neural Network should be done in Introduction section so that Problem Statement can be elaborated accordingly in Section 3.

Apart from other minor mistakes in terms of spelling and grammar line 215 and 216 contains typo errors. Table on results associated with figure 10 is neither labeled nor cited anywhere.

Experimental design

In line number 73 authors claim that this is the first research on forged money using GANs. This statement needs to be proved in light of following work:
Generative adversarial nets by I Goodfellow, J Pouget-Abadie, M Mirza and available at https://papers.nips.cc/paper/5423-generative-adversarial-nets . This paper is published in 2014 and many advances have been made by now (requires evaluation). Moreover GANs has been extensively used in detecting fake object. Such works should also be reported in related work section

The authors should also report about the unique features in Pakistani Currency that makes it unique from other countries. Otherwise just because it is a refined area and since no one did it for Pakistani currency does not seem to be valid hypothesis.

Validity of the findings

Data Preparation and Augmentation section is extremely confusing. The whole section needs revision for better readability. The discussion on function calling and parameters (see line 202-204) is made in haphazard way. A pictorial representation of program flow may better be used to show its correctness.
Tabular results are not shown properly. It is not shown how Confusion Matrix is calculated.

Figure 13 - 15 are showing the results as graphs. However these graphs lack the visiblity issues. Instead the authors should show in results how (for example 80% classification accuracy) was achieved. The screenshots of Tensorflow Graph Visualization is confusing and better methodology should be adapted for reporting the results

Additional comments

Apart from my comments mentioned in above review, various coloring schemes are used to explain the paper. From visualization purpose it is good but in print it may be represented on gray scale and thus make it useless (for example confusion matrix). You may need to check with the Editors in this regard

Reviewer 3 ·

Basic reporting

1. It is recommended to bring the nearest and state-of-the-art works in a tabular form and place it in the “related work” section by identifying purpose, methods, limitations, results etc. of each study.
2. Please extend the proposed scheme to bring more novelty and contributions.
3. Tell me the connection of the proposed work with the state of art techniques for big-data. Besides, the literature is limited.

Experimental design

4. Pose research questions and try to find their answers with experiments.
5. The experimentation and implementation section is weak, where authors should give a detailed experimental setup and find answers to their posed research questions.

6. The pseudo code can be represented to depict the methodology steps followed.

Validity of the findings

8. The results and discussion section should contain both quantitative and qualitative discussion. I want to see some innovative findings out of your implemented work.

Additional comments

Written English must be improved as this does affect the overall understanding of the manuscript, there are many mistakes, equations without numbering, and many long sentences/paragraphs without references
The conclusion and abstract sections should be revised and synchronized accordingly.

---

## Round 0.2 · Minor Revisions

Please improve language quality. You should check this -- if possible -- with a native speaker and/or a language professional. Please check also additional details (see e.g. section 0.1 -- 0.11) in the manuscript format.

[]

Reviewer 3 ·

Basic reporting

Paper is refined

Experimental design

Experiments are well presented now

Validity of the findings

findings are now of sufficient standard to be accepted

Additional comments

All required changes incorporated

---

## Round 0.3 · accepted · Accept

Thank you for your revisions.

Reviewer 4 ·

Basic reporting

N/A

Experimental design

N/A

Validity of the findings

N/A

Additional comments

The paper has been revised well.